# Prevalence of Physical Activity and Sedentary Behaviors in the French Population: Results and Evolution between Two Cross-Sectional Population-Based Studies, 2006 and 2016

**DOI:** 10.3390/ijerph19042164

**Published:** 2022-02-15

**Authors:** Charlotte Verdot, Benoît Salanave, Salomé Aubert, Andréa Ramirez Varela, Valérie Deschamps

**Affiliations:** 1Nutrition Surveillance and Epidemiology Team, Santé Publique France, Centre of Research in Epidemiology and Statistics, University Sorbonne Paris Nord, F-93017 Bobigny, France; benoit.salanave@univ-paris13.fr (B.S.); valerie.deschamps@univ-paris13.fr (V.D.); 2Active Healthy Kids Global Alliance, Ottawa, ON K1H 5B2, Canada; salome_aubert@hotmail.fr; 3School of Medicine, Universidad de los Andes, Bogota 11001000, Colombia; aravamd@gmail.com

**Keywords:** physical activity, sedentary behavior, screen time, recommendations, prevalence, epidemiology, adults, children, France, population-based cross-sectional study

## Abstract

Insufficient physical activity and sedentary behaviors (SB) are major risk factors for non-communicable diseases. Monitoring the prevalence of physical activity (PA) and SB is essential to meet the health needs of the population. This article presents the prevalence of PA and SB in the French population and their evolution during the last decade. Data come from two cross-sectional surveys, representative of the population in France, the “Etude Nationale Nutrition Santé” 2006–2007 and the Esteban study 2014–2016, and were collected through the International Physical Activity Questionnaire and the Recent Physical Activity Questionnaire for adults, the Youth Risk Behavior Survey and specific questionnaire for children. In 2014–2016, 71% of men and 53% of women met the PA recommendations (5 or more days per week with a moderate-intensity physical activity of at least 30 min per day). Since 2006–2007, PA has decreased for women, but increased for men; 80% of adults reported a daily leisure screen time of at least three hours in 2014–2016, in strong growth since 2006–2007. Among children, only 51% of boys and 33% of girls were meeting the PA recommendations (at least 60 min of moderate- to vigorous-intensity physical activity daily). PA decreased significantly after the age of 10. Three-quarters of children spent two hours or more in front of a screen every day. These results show a lack of PA, in particular among women and girls, a high prevalence of SB in the French population, and a deterioration of these behaviours between 2006 and 2016.

## 1. Introduction

Physical activity is defined as “*any bodily movement produced by skeletal muscles that requires energy expenditure*” [1]. It included physical activity at work, at home, for transport, and during leisure time. Any physical activity has significant health benefits for all age categories [2,3,4,5]. Being physically active is also recommended to maintain or improve cardiorespiratory and functional health and to limit the risk of non-communicable diseases (NCDs), such as cardiovascular diseases, obesity, cancer, or diabetes [1,3,6,7]. The current WHO recommendations on physical activity for health in children and adolescents (5–17 years) are to participate in at least 60 min of moderate to vigorous-intensity physical activity daily, and for adults, at least 150–300 min of moderate-intensity physical activity throughout the week, or at least 75–150 min of vigorous-intensity physical activity, or an equivalent combination of moderate- and vigorous-intensity activity [8]. In France, guidelines recommended that adults spread out their activity during the week, in doing at least 30 min of moderate-intensity physical activity on at least five days, or at least 25 min of vigorous-intensity physical activity on at least three days, or equivalent. 

In 2009, insufficient physical activity (defined as the non-achievement of the WHO recommendations) was identified as the fourth risk factor for NCDs, implicated in more than 3 million preventable deaths [9]. Insufficient physical activity is responsible for 6–9% of all-cause mortality worldwide, and 6% of coronary heart diseases, 7% of type 2 diabetes, 10% of breast cancer, 10% of colon cancer, and 9% of premature deaths are due to being physically inactive [6]. Moreover, insufficient physical activity is even more detrimental when it is coupled with a substantial sedentary lifestyle. A sedentary behavior is defined by any behavior characterized by an energy expenditure ≤ 1.5 METs (metabolic equivalent task) while in a sitting, reclining, or lying posture [10]. In the literature, sedentary behaviors are approached by various indicators or proxies, such as the daily sitting time or the daily screen time (television or computer use on leisure time) [11]. All are associated with increased cardio metabolic risks and all-cause mortality in case of increasing daily durations [12,13,14,15]. 

To obtain substantial health benefits, individuals must be physically active and limit their sedentary behaviors. However, Guthold et al. estimated that the global prevalence of insufficient physical activity in 2016 was 27.5%% among adults aged 18 and older (36.8% in high-income western countries) [16] and 81.0% among adolescents aged 11–17 years [17]. In addition, 18.5% of European adults would spend more than 7.5 h per day sitting (with a median of 5 h per day) [18] and two thirds of 13–15 year old would watch two or more hours of television each day [19]. In contrast, in China, only a quarter of school-age children do not meet screen time guidelines [20].

Given the situation, it is therefore urgent to limit insufficient physical activity and sedentary behaviors and to promote active lifestyles in order to prevent NCDs and to limit their morbidity [12,15,21]. Monitoring the prevalence of physical activity and sedentary behaviors is also essential to accurately assess the situation, to identify needs, to track evolutions, and to define and adapt programs and actions to implement [22]. The prevalence of the physical activity and screen time during leisure (as well as their adequacy with health recommendations) were thus measured in France in 2006, as part of the National Health and Nutrition Survey (Etude Nationale Nutrition Santé, ENNS 2006–2007), and more recently in 2014–2016, as part of the Esteban study (Health Study on Environmental, Biomonitoring, Physical Activity and Nutrition). These two national studies, conducted by *Santé publique France* (i.e., the French public health agency), produced representative data on the French population. The aim of this article is to present the prevalence of physical activity and leisure screen time of adults and children living in France, in 2006 and 2016.

## 2. Materials and Methods

### 2.1. Study Design

In France, the national nutritional surveillance includes monitoring dietary intake, physical activity prevalence, sedentary behaviors, and nutritional status of the population. This nutritional surveillance is carried out by nationwide cross-sectional studies, with a multi-stage sampling design: i.e., random selection of geographic zones (stratified on regions and degree of urbanization); then, random selection of households based on randomly generated phone lists; finally, random selection of eligible subjects. Recruitment was carried out throughout the entire territory of continental France for one year so as to account for seasonality. The study protocol includes a questionnaire survey (face-to-face and self-administered questionnaires), a diet survey (three 24 h dietary recalls), and a health examination with clinical and biochemical measurements.

Two cross-sectional population-based studies have been carried out in recent years: ENNS in 2006–2007 and Esteban in 2014–2016. The design and methods (already described in detail elsewhere [23,24]) were similar between the two studies, in order to have comparable data to study evolution of the indicators over time.

These studies received the approval of the Advisory Committee on Information Treatment in the field of Health Research (CCTIRS), the French Data Protection Authority (Cnil), and the Personal Protection Committee (CPP). All participants signed informed consents.

### 2.2. Sociodemographic Data

For both studies (ENNS and Esteban), social and demographic information was collected through face-to-face questionnaires. These data included the sex, age, and education level (highest degree obtained) of the participants (or of the adult responsible for the household in the case of children), as well as general question, such as the average time spent outdoors for children.

### 2.3. Measures of Physical Activity and Sedentary Behaviors

Data were collected by self-questionnaires adapted to the age of the participants.

In ENNS, adults completed the short form of the International Physical Activity Questionnaire (IPAQ), which records the activity of different intensity levels (vigorous-intensity, moderate-intensity, walking and sitting time) over the past seven days [25]. Questions on the average daily leisure screen time (including TV, computer, and game console time outside of any professional or educational activity) were added to complete sedentary data. In Esteban, adults completed the Recent Physical Activity Questionnaire (RPAQ), which is more detailed and assess daily physical activity over the past four weeks detailed by type of activities (sports, transport, leisure…) [26]. Home activities (cleaning, gardening, and DIY) have been added in the RPAQ for comparison with the IPAQ and to include the measure of physical activity in all areas of daily life (at work, at home, for transport, and during leisure time). The comparison of data between the IPAQ and the RPAQ was based on the estimation of common indicators, i.e., weekly energy expenditure (expressed in METs-minutes/week) and the number of days per week with at least 30 min of moderate-intensity physical activity (3.0–5.9 METs) or at least 25 min of vigorous-intensity physical activity (≥6.0 METs).

In 2006, as part of the ENNS, physical activity and sedentary behaviors of adolescents aged 15–17 years were measured using the IPAQ, as for adults. For children aged 11–14, the Youth Risk Behavior Survey (YRBS) was used with an adaptation of items to the French recommendation of 60 min of physical activity per day. In the Esteban study, the use of this tool was extended to children and adolescents aged 11–17. Items assessed moderate- and vigorous-intensity physical activity, strengthening exercises, physical education at school, participation in sport clubs, and screen time in the last seven days. Both questionnaires, IPAQ and YRBS, describe global physical activities over the past seven days and by level of intensity (vigorous, moderate, …). However, IPAQ records detailed durations per day and the number of days per week, whereas the YRBSS only collects the number per week of days during which physical activity reached the recommended level.

Finally, for children aged 6–10, a specific questionnaire (the same in the two studies) was filled out by the parents, with questions on physical activity at school, active play in outdoor, participation in sport clubs, active transportation to school, and screen time in the last seven days. As the general information collected from children being on vacation during the week surveyed was not similar between ENNS and Esteban (concerning the average time spent outdoors), evolutions analysis between 2006 and 2016 only included children who went to school during the week interviewed (for whom the same questions were asked in both surveys).

### 2.4. Data Analysis

The prevalence of physical activity was calculated according to current French health recommendations (for adults, at least 30 min of moderate-intensity physical activity on at least five days, or at least 25 min of vigorous-intensity physical activity on at least three days, or equivalent; and for adolescents and children, at least 60 min of moderate- to vigorous-intensity physical activity daily) [8]. The construction of this indicator is detailed for each age group in Table 1.

Time spent sedentary was estimated using daily leisure screen time. We reported the percentage of adults reporting a daily leisure screen time of three hours or more, and that of children spending two hours or more on screens each day [27].

The prevalence of physical activity and daily leisure screen time were calculated using a comparable method for the two studies (ENNS and Esteban), in order to be able to discuss comparisons between 2006 and 2016. Prevalence estimates included 95% confidence intervals (95% CI) and were presented with stratification by sex. All differences between sex, age, education level, and between the two surveys were examined by Pearson’s Chi-square test corrected with the Rao-Scott method.

All analyses were performed on weighted and calibrated data using Stata14^®^ software. Sampling weights were calculated according to the sampling design and calibrated on census data (sex, age, education level, presence of at least one child in the household). The complex design of the study was considered particularly in the estimation of variances and 95% CI using the “svyset” function in Stata.

The statistically significant changes were confirmed by standardization of the ENNS data on the characteristics of the Esteban population, in order to avoid any effect due to the possible change in the profile of the population during the period. This standardization of ENNS data was achieved by calculating a new set of weights for ENNS, according to the same adjustment principles and with the same calibration data (on sex, age, education level) as those used for the Esteban calibration.

## 3. Results

In the Esteban study, analyses of physical activity and sedentary behaviors included 2682 adults aged 18–74 years and 1281 children aged 6–17 years for whom data were available (Figure 1). In ENNS, analyses included 2971 adults and 1358 children of the same age [24].

### 3.1. Prevalence of Physical Activity and Sedentary Behaviors in Adults in the Esteban Study (2014–2016)

In 2014–2016, 70.6% [67.0–73.9] of men and 52.7% [49.3–56.1] of women met the recommendations on physical activity for health (significant gender difference; *p* < 0.001). Analyses conducted by age groups or education levels (based on the highest diploma obtained) showed no difference in the prevalence of physically active adults (Table 2). However, this prevalence was lower among people who reported a daily leisure screen time greater than or equal to three hours (significantly among men *p* < 0.01; not among women *p* = 0.08). The lower prevalence of physical activity for women compared to men was statistically significant regardless of age, education level, or screen time.

Concerning sedentary behaviours, 80.5% [77.4–83.2] of men and 79.8% [77.1–82.2] of women reported a daily leisure screen time greater than or equal to three hours (Table 2). There were no significant differences by gender, regardless of age, education level, or achievement of physical activity recommendations. However, this prevalence was lower among women aged 40–54 (*p* < 0.01), among people with a level of education at least equal to a master’s degree, for both men and women (*p* < 0.001), and among men who met the recommendations on physical activity (*p* < 0.01).

### 3.2. Physical Activity Prevalence of Adults since 2006

In 2006–2007, the prevalence of physical activity was similar for men and women (i.e., 63.2% [60.8–65.5]), which was no longer the case in 2014–2016. Between 2006–2007 and 2014–2016, the proportion of physically active men increased by 10% (*p* < 0.05), while the proportion of physically active women fell by almost 16% (*p* < 0.001). Men aged 40–54 experienced a significant change in their physical activity: in 10 years, the proportion of men reaching the recommendations increased by 30% in this age group (*p* < 0.01; Figure 2). Conversely, for women, the decrease in physical activity was observed in all age groups. There was no change in the prevalence of physical activity according to individuals’ education level.

### 3.3. Daily Leisure Screen Time of Adults since 2006

In 2006–2007, the prevalence of adults reporting three or more hours of daily leisure screen time was 53.2% [50.8–55.7], while it reached 80.1% [78.1–82.0] in 2014–2016. This increase was more pronounced in women (+67%; *p* < 0.001) than in men (+37%; *p* < 0.001). This was statistically significant across all age groups (*p* < 0.001; Figure 3). It was highest among women aged 40–54 years (+113% between the two studies; *p* < 0.001).

While the daily leisure screen time has increased over 10 years for the entire adult population and for all education levels (*p* < 0.001), this increase was greater among the least educated (+42% for men and +74% for women with less than a high school degree; *p* < 0.001) and among adults with a bachelor’s degree (+39% for men and +124% for women; *p* < 0.001).

### 3.4. Prevalence of Physical Activity and Sedentary Behaviors in Children in the Esteban Study (2014–2016)

In 2014–2016, 50.7% [45.1–56.3] of boys and 33.3% [28.4–38.6] of girls aged 6–17 met the physical activity recommendations (significant gender difference; *p* < 0.001). Younger children (6–10 years old) were more likely to achieve adequate physical activity prevalence (*p* < 0.001), as were boys compared to girls in each age group and whatever daily screen time (Table 3).

There was a marked decrease in physical activity after age 10, with a greater decrease for girls and a continuing decline as they got older. The prevalence of children achieving 60 min of physical activity per day tended to increase with the education level of the responsible adult; however, this was not globally statistically significant. On the other hand, this prevalence was higher among children spending less than 2 h a day in front of the screen than the others (*p* < 0.01), regardless of gender (Table 3).

The physical activity of the 6–10 year olds was mostly due to school-based physical education, active play outdoor, and participation in sport clubs. Less than half (40.2% [33.7–47.1]) reported an active mode of transportation (walking, cycling, scooter, rollerblades) to go to school. The physical activity of children aged 11–14 year olds was mostly due to school-based physical education and participation in sport clubs, which was more common among children from the most highly educated households. Finally, the physical activity of 15–17 year olds was the result of school physical education, participation in sport clubs (also more frequent in the most educated households but decreasing compared to 11–14-year-olds), and weight training exercises that were more popular among boys. 

Concerning sedentary behaviors, 80.7% [75.8–84.8] of boys and 73.4% [68.2–78.0] of girls aged 6–17 reported a daily screen time greater than or equal to two hours (significant gender difference; *p* < 0.05; Table 3). This percentage increased with age (*p* < 0.001), decreased with the education level of the responsible adult (*p* < 0.05) and was lower among children achieving 60 min of physical activity per day (*p* < 0.01), regardless of gender.

### 3.5. Physical Activity Prevalence of Children since 2006

The percentage of children and adolescents reaching the recommended 60 min of physical activity per day has not changed significantly over the past 10 years, regardless of sex and age group. There is, however, an increase in the prevalence of physical activity among boys aged 15–17 years, but this is not statistically significant (Figure 4).

### 3.6. Daily Leisure Screen Time of Children since 2006

The percentage of children and adolescents reporting a daily screen time greater than or equal to two hours has increased by 17% in 10 years (it was 65.5% [62.2–68.7] in 2006–2007 versus 76.9% [73.3–80.1] in 2014–2016; *p* < 0.001). This increase was generalized to all children, regardless of the gender, age, or education level of the responsible adult. It was highest among boys aged 6–10 years (+23%; *p* < 0.01), boys aged 15–17 years (+24% *p* < 0.001), and girls aged 15–17 years (+26% *p* < 0.01) (Figure 5).

## 4. Discussion

This is the first article to present the estimation of the prevalence of physical activity and sedentary behaviors of the French population using national data (Esteban 2014–2016) across children, adolescent, and adult age groups, and their association with age and level of education. It is also the first article to compare these behaviors from two time-points over 10 years.

The results of the Esteban study indicate that in 2014–2016, 61.4% of adults living in metropolitan France met the physical activity recommendations. This rate was lower than the average rate in high-income western countries (63.2% according to the study of Guthold et al. based on the same recommendations [16]), but higher than those of 51.6% and 52.6% observed in the United States and Australia, respectively [28,29].

Furthermore, women were less physically active than men, with only 53% of women meeting the physical activity recommendations compared to 71% of men. The prevalence of physically inactive men, around 29% in France, is below the average for high-income western countries (31.2%), while that of women is almost five points higher (47% in France versus 42.3% on average in high-income western countries [16]). A lower prevalence of physical activity was generally observed among women in population surveys, but this difference has been particularly marked in recent years in France. The low prevalence of physical activity among women is concerning, given the decrease in this indicator in recent years. Between 2006–2007 and 2014–2016, the proportion of women reaching the physical activity recommendations fell by almost 16%. This decline was more pronounced among women aged 40–54 (−21%), while, conversely, among men in the same age group, there was a significant increase in the prevalence of physical activity (+30% over the last 10 years). To our knowledge, no other population-based studies have observed such divergent evolution of physical activity behaviors between men and women in recent years. The next French survey, scheduled in 2024 and including a new RPAQ measurement, would confirm these trends and analyze more precisely the nature of French adults’ practices to explain these evolutions. In the meantime, this highlights the importance of taking these differences into account when developing prevention actions for the general population. 

Among children and adolescents, the results of the Esteban study showed that in 2014–2016, only half of boys and one third of girls aged 6–17 years met the physical activity recommendations. This rate varied according to gender and age. Boys were more active than girls and physical activity prevalence decreased significantly after the age of 10. Although quite low, these figures were nevertheless higher than those found in the “Health Behaviour in School-aged Children” study, which reported an international prevalence averaging 28% and 19% for boys aged 11 and 15 and 19% and 10%, respectively, for girls of the same age [19]. 

The percentage of children and adolescents reaching physical activity recommendations has not changed significantly over the past 10 years. However, results showed an increase in the overall prevalence of physical activity among 15–17-year-olds, as observed in Europe and North America between 2002 and 2010 [30]. These data highlight the need for appropriate physical activity promotion actions targeting each age group to maintain adequate prevalence of physical activity among the youngest age groups and to confirm the observed increase in the overall prevalence of physical activity among the oldest age groups.

While children’s involvement in leisure-time physical activity remains, according to literature and confirmed by our study, dependent on many social inequalities (i.e., household income, parental education level, parental employment, ethnic minorities, and availability of physical recreation and sporting facilities) [31,32,33], increasing time dedicated to school-based physical education in the school curriculum could be a strategic way to improve access to regular physical activity for all children and adolescents, regardless of age, gender, or socio-economic level, as well as encourage the development of active outdoor play [34]. It also seems important to further increase active transportation to go to school, given the low rate of children and adolescents with active mobility (only 40% of 6–10 year-olds in the Esteban study) and the positive impact of this behavior on children’s health [35].

Regarding the prevalence of sedentary behaviors in the French population, the Esteban study also highlighted the high levels of screen time in the French population: in 2014–2016, 80% of adults declared spending more than three hours a day in front of a screen outside their professional activity and 77% of children spent two hours or more in front of a screen each day. This prevalence is higher than the international estimates reporting two-thirds of the children with screen time of 2 h or more [34], and 64% among 6–7 aged Italian children [36]. Screen time was also higher among adults with lowest education level and among children from the least educated households, reflecting, as in international studies, the presence of social inequalities in the development of this behavior [37]. For adults, the threshold of 3 h per day complicates comparisons with other studies. This threshold has been retained here because it easily allows the comparison between ENNS and Esteban. However, this threshold often varies from one study to another and the indicators of sedentary lifestyle often relate to different items (sitting time, leisure screen time, screen time including professional or educational activities, …). Moreover, the use of screens evolves according to their types (television, computer, video games, videos, smartphones) making comparisons over time sometimes difficult. The type of screen used can also have different effects on the health of populations. Indeed, looking at your smartphones in transport does not have the same impact on your health as spending hours sitting in an armchair watching television.

Daily leisure screen time has strongly increased during the last decade for the entire population and more pronounced among women, boys aged 6–10, and youth aged 15–17. Preventive actions must therefore be directed mainly towards these groups and take into account the evolution and development of screen uses (computers, smartphones) to limit the sedentary behavior of the population. Moreover, among children and adolescents, several studies have shown that the increase in time spent outdoors, particularly in informal after-school activities, is associated both with better adherence to physical activity recommendations and with the limitation of sedentary behavior, including screen time [34,37,38]. Spending more time outdoors should therefore be encouraged, either through specific programs or through the development of supportive environments (playgrounds, parks, etc.).

When considering insufficient physical activity and sedentary behavior as risk factors for health in both children and adults, the opportunities and benefits of promoting active systems to build better societies are evident. Physical activity contributions should not be limited to health outcomes, and should include different sectors of the society and consider social, cultural, environmental, and economic factors to reverse insufficient physical activity and reduce inequalities. Specifically, the Global Action Plan on Physical Activity (GAPPA) launched by WHO in 2018 represents a guide to countries as it provides a framework for delivering policy actions for supporting systems to increase physical activity [22]. The GAPPA can be used together with the recently launched 2020 WHO guidelines to continue improving the French surveillance system and policies to promote physical activity and achieve the global targets [39,40]. 

### Strengths and Limitations

The comparative data between ENNS and Esteban studies should be considered with caution given the changes in the tools used to measure physical activity in the population (RPAQ versus IPAQ in adults, YRBS versus IPAQ in adolescents, and the addition of questions on weight training and sports club attendance in 11–14-year-olds). These methodological changes may have affected the accuracy of the data, particularly the precise definition of activity intensity. However, these are two population-based surveys and the study of common indicators, calculated as close as possible between the two studies, made it possible to estimate the general changes that have taken place over the last 10 years. Like most studies, ENNS and Esteban may have been affected by the classic biases of self-administered and face-to-face questionnaires. Moreover, concerning children under 11, parents were asked to answer on behalf of their children. However, without being able to assess these biases, these biases are supposed to be limited and nothing can suggest that they could have evolved significantly over this period. The next wave of data collection will have to repeat these observations to measure these behaviors and study their evolution over time, but it will also have to produce additional data (such as active transportation or social environment). The aim is to respond to the set of indicators retained in international monitoring systems, such as the Active Healthy Kids Global Alliance Report Cards on physical activity for children and youth [27,41].

From 2006 to 2016, the prevalence of meeting physical activity recommendations decreased among women. Over the same period, screen time has increased significantly, regardless of age and gender. Additionally, the deterioration of these indicators cannot be explained by the evolution of the characteristics of the population between the two studies (as shown by the standardization).

Due to the COVID-19 pandemic, this surveillance data is critical and has a strong role in global physical activity advocacy. The indicators can be used directly and immediately to influence the physical activity agenda. It is important to monitor the effects of the COVID-19 pandemic on physical activity and vice versa. Insufficient physical activity is one of the biggest risk factors for severe COVID-19 outcomes. Good public health practice must be promoted.

## 5. Conclusions

The results of the Esteban study provide an overview of the physical activity and sedentary behaviors of adults and children living in metropolitan France in 2014–2016. Comparisons with the data collected in 2006–2007 in the ENNS study allowed the study of the evolution of these indicators over the last 10 years.

The low prevalence of physical activity and high sedentary lifestyle were reported among adults and children living in metropolitan France in 2014–2016, and a decline in these indicators was observed over 10 years (2006–2007 to 2014–2016). These results highlight the need for the development of effective interventions and actions to: (1) increase the population’s prevalence of physical activity; and (2) limit the time spent in sedentary behaviors. It is necessary to act on these two factors independently of each other and in a targeted manner according to individual’s needs. Particular attention must be paid to women, among whom these factors deteriorated more markedly over the last 10 years, and to the need to reduce the social inequalities still present in terms of physical activity and even more so in terms of sedentary behaviors.

## Figures and Tables

**Figure 1 ijerph-19-02164-f001:**
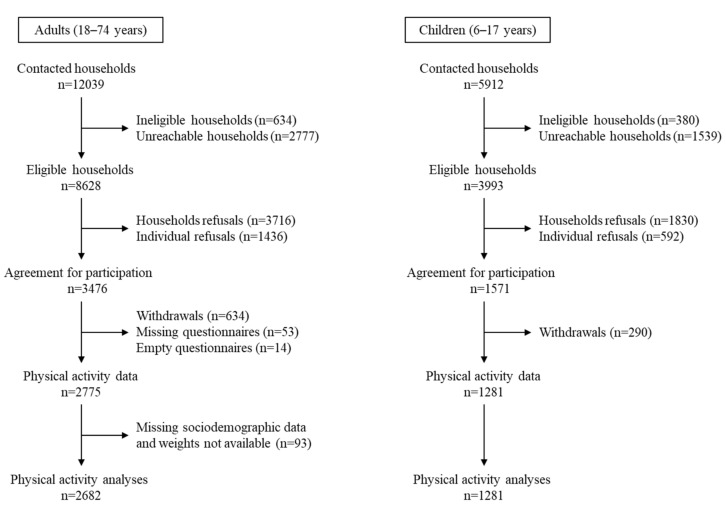
Flow diagram of inclusion—Esteban study (2014–2016).

**Figure 2 ijerph-19-02164-f002:**
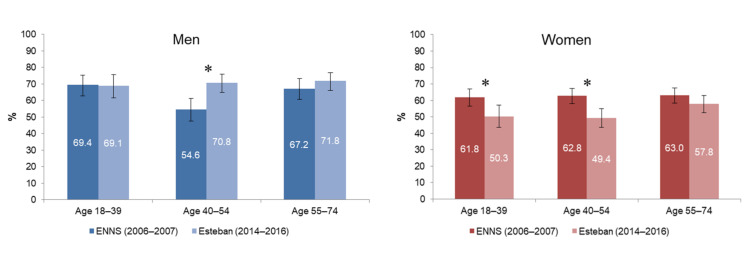
Prevalence of physical activity in men and women aged 18–74 years, between ENNS (2006–2007) and Esteban (2014–2016). * significant change *(p* < 0.01).

**Figure 3 ijerph-19-02164-f003:**
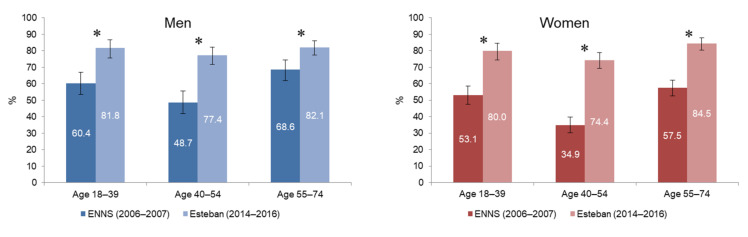
Prevalence of men and women aged 18–74 years reporting 3 or more hours of daily leisure screen time ^1^, between ENNS (2006–2007) and Esteban (2014–2016). ^1^ Leisure screen time includes TV, computer and game console time outside of any professional activity. * significant change (*p* < 0.001).

**Figure 4 ijerph-19-02164-f004:**
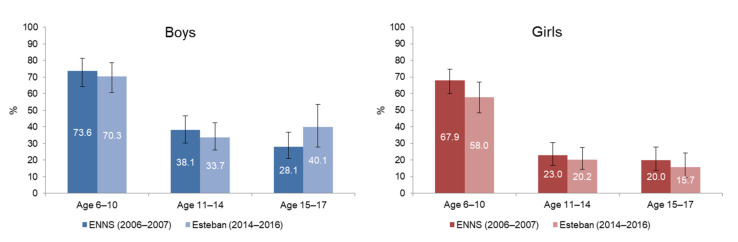
Prevalence of physical activity in boys and girls aged 6–17 years, between ENNS (2006–2007) and Esteban (2014–2016).

**Figure 5 ijerph-19-02164-f005:**
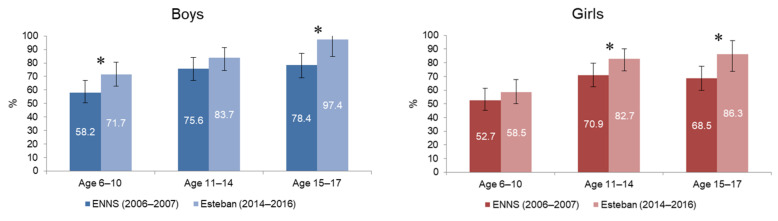
Prevalence of boys and girls aged 6–17 years spending two hours or more on screens ^1^ each day, between ENNS (2006–2007) and Esteban (2014–2016). ^1^ Screen time accumulates TV, computer and game console time. * significant change (*p* < 0.01).

**Table 1 ijerph-19-02164-t001:** Level of physical activity required to meet health recommendations for adults and children.

Age Group	Level of Physical Activity Required
Adults 18 –74-year-olds	Accumulate 3 or more days per week with a vigorous-intensity physical activity of at least 25 min per day; Or accumulate 5 or more days per week with a moderate-intensity physical activity of at least 30 min per day; Or accumulate 5 or more days per week with a moderate- to vigorous-intensity physical activity to reach a minimum of 600 METs per week.
Adolescents 11–17-year-olds	Accumulate at least 60 min of moderate- to vigorous-intensity physical activity daily; Or accumulate 5 or more times per week with a vigorous-intensity physical activity of at least 40 consecutive minutes.
Children 6–10-year-olds	For children at school during the week surveyed: Accumulate 5 or more days per week with physical activity (including active play in outdoor, organized sports, physical education and active transportation to get to school). For children on holiday during the week surveyed: Child considered “physically active” by his parents and 3 or more days per week with organized sports; Or child considered “physically active” by his parents and average daily time spent outside ≥90 min ^1^.

^1^ assumption that the child is active 60 min out of the 90 min spent outdoors.

**Table 2 ijerph-19-02164-t002:** Prevalence of physical activity and sedentary behaviors in adults (18–74 years old) in the Esteban study (2014–2016).

	Men (*n* = 1169)		Women (*n* = 1513)		*p* ^1^
	%	95%CI	%	95%CI	
**Prevalence of physical activity**(achievement of recommendations)	70.6	[67.0–73.9]	52.7	[49.3–56.1]	<0.001
Age groups					
18–39 years old	69.1	[61.7–75.5]	50.3	[43.6–57.0]	<0.001
40–54 years old	70.8	[64.9–76.0]	49.4	[43.7–55.1]	<0.001
55–74 years old	71.8	[67.0–73.9]	57.8	[52.6–62.9]	<0.001
Education level					
<High school degree	70.9	[65.0–76.1]	51.6	[45.9–57.2]	<0.001
High school degree	69.7	[60.9–77.2]	54	[46.9–60.9]	0.006
Bachelor’s degree	74.6	[67.4–80.6]	50.2	[43.6–56.8]	<0.001
Master’s degree	68.5	[61.9–74.3]	57.3	[50.8–63.5]	0.02
Daily leisure screen time					
<3 h/day	**80.1**	**[73.5–85.4]**	58.3	[51.3–65.0]	<0.001
≥3 h/day	**68.3**	**[64.2–72.1]**	51.3	[47.4–55.2]	<0.001
**Prevalence of sedentary behaviors** (daily leisure screen time ≥3 h/day)	80.5	[77.4–83.2]	79.8	[77.1–82.2]	0.7
Age groups					
18–39 years old	81.8	[75.6–86.7]	**80**	**[74.6–84.5]**	0.6
40–54 years old	77.4	[71.9–82.1]	**74.4**	**[69.5–78.9]**	0.4
55–74 years old	82.1	[77.3–86.0]	**84.5**	**[80.5–87.8]**	0.4
Education level					
<High school degree	**86.3**	**[81.6–89.9]**	**85.3**	**[80.9–88.7]**	0.7
High school degree	**77.1**	**[68.5–83.9]**	**81.2**	**[75.1–86.1]**	0.4
Bachelor’s degree	**80.9**	**[74.1–86.2]**	**78.6**	**[73.2–83.2]**	0.6
Master’s degree	**69**	**[62.8–74.5]**	**62.1**	**[55.8–68.1]**	0.1
Recommendations on physical activity					
achieved	**77.8**	**[73.9–81.3]**	77.6	[73.9–81.0]	0.9
non-achieved	**86.8**	**[82.1–90.4]**	82.2	[78.2–85.6]	0.1

^1^ *p* value for the difference between men and women. In bold, significant difference in PA prevalence by screen time for men (*p* < 0.01), significant difference in sedentary prevalence: by age group for women (*p* < 0.01), by education level for men and women (*p* < 0.001) and by achievement of PA recommendations for men (*p* < 0.01).

**Table 3 ijerph-19-02164-t003:** Prevalence of physical activity and sedentary behaviors in children (6–17 years old) in the Esteban study (2014–2016).

	Boys (*n* = 643)		Girls (*n* = 638)		*p* ^1^
	%	95%CI	%	95%CI	
**Prevalence of physical activity** (achievement of recommendations)	50.7	[45.1–56.3]	33.3	[28.4–38.6]	<0.001
Age groups					
6–10 years old	**69.7**	**[61.1–77.1]**	**55.5**	**[47.0–63.7]**	0.02
11–14 years old	**33.7**	**[26.0–42.4]**	**20.2**	**[14.3–27.7]**	0.01
15–17 years old	**40.1**	**[28.0–53.6]**	**15.7**	**[9.7–24.4]**	<0.001
Parent education level					
<High school degree	48.8	[39.7–57.9]	27.7	[20.3–36.5]	0.001
High school degree	47.6	[35.8–59.7]	35.8	[24.3–49.2]	0.2
Bachelor’s degree	56.9	[47.0–66.3]	36	[27.6–45.3]	0.003
Master’s degree	55.2	[45.5–64.6]	42.5	[32.7–53.0]	0.08
Daily screen time					
<2 h/day	**70.3**	**[58.2–80.1]**	**46.7**	**[36.3–57.3]**	0.004
≥2 h/day	**47.6**	**[41.0–54.4]**	**30.1**	**[24.5–36.5]**	<0.001
**Prevalence of sedentary behaviors** (daily screen time ≥2 h/day)	80.7	[75.8–84.8]	73.4	[68.2–78.0]	0.03
Age groups					
6–10 years old	**71.7**	**[63.4–78.7]**	**58.5**	**[49.8–66.8]**	0.03
11–14 years old	**83.7**	**[75.6–89.5]**	**82.7**	**[75.6–88.0]**	0.83
15–17 years old	**97.4**	**[92.7–99.1]**	**86.3**	**[75.6–92.7]**	0.003
Parent education level					
<High school degree	**85.1**	**[76.3–90.9]**	**79.1**	**[70.1–85.9]**	0.28
High school degree	**84.6**	**[73.8–91.5]**	**70.5**	**[57.2–81.0]**	0.06
Bachelor’s degree	**75**	**[65.5–82.6]**	**73.2**	**[63.1–81.4]**	0.78
Master’s degree	**69.4**	**[60.0–77.4]**	**59.6**	**[48.7–69.6]**	0.16
Recommendations on physical activity					
achieved	**73.9**	**[66.2–80.3]**	**64**	**[54.7–72.5]**	0.089
non-achieved	**88**	**[82.2–92.1]**	**78.3**	**[72.1–83.5]**	<0.014

^1^ *p* value for the difference between boys and girls. In bold, significant difference by age group (*p* < 0.001), by parent education level (*p* < 0.05, only for the prevalence of sedentary behaviors), by screen time (*p* < 0.01) and by achievement of PA recommendations (*p* < 0.01), for boys and girls.

## Data Availability

The data is not deposited in publicly available repositories but are available on request from Santé publique France. For more information, please consult: https://www.santepubliquefrance.fr/les-actualites/2019/appel-a-projet-d-ouverture-des-bases-de-donnees-y-compris-de-la-collection-biologique-de-l-etude-esteban-2014-2016. Accessed on: 11 February 2022.

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
