# Peer review of "Prevalence of Physical Activity and Sedentary Behaviors in the French Population: Results and Evolution between Two Cross-Sectional Population-Based Studies, 2006 and 2016"

_ijerph, 2022, doi:10.3390/ijerph19042164_

Round 1

Reviewer 1 Report

Dear authors:

Attached are some aspects of the articles that could be improved:

Abstract

-Description of abbreviations are required (IPAQ, ENNS, RPAQ)

-A briefly description of the PA and screen time recommendations could be helpful.

-Are the cross-sectional surveys (ENNS, Esteban) representative of France population?  

Introduction

-Are to participate IN at least

-In France, guidelines recommended that adults to spread out their activity during the week, in doing at least 30 min of moderate-intensity physical activity ON AT LEAST à to many AT LEAST in the same paragraph.

-You should describe sufficient physically active and insufficient physically active before this paragraph (In 2009, insufficient physical activity…)

-It is quite confusing to read in the first paragraph “physical activity” in the following “physical inactivity” and in the third “physical activity”. I recommend to start describing physical inactivity and then physical activity or vice versa.

-Within introduction section, you mentioned 2 different cuts of points of physical activity prevalence (WHO and France). Are the Guthold estimates based on WHO or France?

-The levels and trends à I recommend to use “prevalence” instead of “levels”

-Sedentary behaviors (are you referring to sitting time or television watching or both?)

Materials and Methods

-What are the main differences between ENNS and Esteban surveys?

- What was the question to estimate the educational level?

-Did you use the short or long form of the IPAQ?

-Are the daily leisure screen time questions validated? If so, could you report this into the methods section?

-What is the main difference between IPAQ and RPAQ? Do you have any information related to validity or mean differences?

-Could you explain in more detail the differences between IPAQ and YRBS?

-Data analysis à prevalence of physical activity was estimated based on France guidelines?

-We reported the percentage of adults reporting a dailyà to similar words in the same sentence.

- What was your rationale behind ≥3 hours/day of leisure screen time cut off point for adults?

-Could you please insert a reference after… “children spending two hours or more on screens each day”

-As I understand, ENNS and Esteban are cross-sectional national representative surveys. Did you use a sample weight? If so, could you explain this in more detail. This could change statistical tests.

-Did you consider any adjustment for age changes among years?

Results

-Could you explain the total number of people that the surveys represented? Is the number similar to what was observed in the entire population for the same year?

-What is the total physical activity prevalence?

-Did you make some adjustment to compared the prevalence of last week (IPAQ) and the last 4 weeks (RPAQ)? Same comment for children and adolescents (IPAQ vs YRBS)

-Could you please include total physical activity prevalence within the figure 2 or within the paragraph?

-Could you also report total physical activity/screen time prevalence for those aged 6-17 years?

Discussion

-You are comparing your prevalence (France physical activity guidelines) vs Guthold prevalence. Did both surveys use the same cut-off point for physical activity? If not, you should add this limitation into the second paragraph.

-Within the third paragraph, you should explain why women had lower prevalence than men and why the trend was markedly lower in women.

-Paragraph 4. You are comparing prevalence estimated using IPAQ and YRBS vs prevalence estimated by HBSC. Please address this issue in the paragraph.

-Is there any strategy that could contribute to increase physical activity levels among 15-17-year olds in France? If so, could you explain in more detail.

-Please correct the following sentence: “this prevalence is higher that the international estimates reporting two-thirds of the children concerned 34.”

-Could you also discuss the screen time cut-off point for adults?

-Could you discuss strategies to reduce screen time in women?

-What about the challenges of increasing PA and reducing screen time during COVID-19?

Limitations

-Could phone surveys biased the entire sample?

-IPAQ collected information in the last week, whereas RPAQ in the last 4 weeks. What did you expected to observed and how did you fix it?

-Parents answered questions related to children’s physical activity, so, please explain limitations related to this.

-Explain other bias related to self-report instruments.

Reviewer 2 Report

The authors (AA) aim to to present the levels and trends of physical activity and sedentary behaviors of adults and children living in France, from 2006 to -2016. This is an engaging article with a study design appropriate and useful to increase our knowledge of the issue. The title reports the key features of the paper encouraging the reader to read more.

Addressing the following issues can make this interesting manuscript eligible for the publication.

Abstract

Line 16: Specify the period (the last decade?)

Introduction

The references are recent, relevant and appropriate. I suggest that AA could add some other references about their topic regard children (6-10 years) as reported below:

⁻              Zhu, X.; Haegele, J.A.; Tang, Y.; Wu, X. Prevalence and Demographic Correlates of Overweight, Physical Activity, and Screen Time Among School-Aged Children in Urban China: The Shanghai Study. Asia-Pacific journal of public health 2018, 30, 353 118-127, doi:10.1177/1010539518754538.

⁻              Paduano S, Borsari L, Salvia C, Arletti S, Tripodi A, Pinca J, Borella P. Risk Factors for Overweight and Obesity in Children Attending the First Year of Primary Schools in Modena, Italy. J Community Health. 2020 Apr;45(2):301-309. doi: 10.1007/s10900-019-00741-7.

⁻              Kumar, S.; Kelly, A.S. Review of Childhood Obesity: From Epidemiology, Etiology, and Comorbidities to Clinical Assessment and Treatment. Mayo Clinic proceedings 2017, 92, 251-265, doi:10.1016/j.mayocp.2016.09.017.

⁻              Paduano S., Greco A., Borsari L., Salvia C., Tancredi S., Pinca J., Midili S., Tripodi A., Borella P., Marchesi I. Physical and Sedentary Activities and Childhood Overweight/Obesity: A Cross-Sectional Study among First-Year Children of Primary Schools in Modena, Italy. Int J Environ Res Public Health. 2021 Mar; 18(6): 3221. doi: 10.3390/ijerph18063221.

Materials and Methods

Did AA consider nationality of subjects for adults or nationality of parents for children?

Results

Did AA perform a comparison of participants’ socio-demographic characteristics between 2006 and 2016? It would important in order to discuss results.

Results are well exposed, figures are appropriate to draw attention to the data. I suggest to add a graph about trends in physical activity levels of children during the decade 2006-2016 (lines 238-242), even if the differences are not statistically significant.
